# Impact of the period of the day on all-cause mortality and major cardiovascular complications after arterial vascular surgeries

Thiago Artioli[1], Danielle Menosi Gualandro[2,3], Francisco Akira Malta Cardozo[2], María Carmen Escalante Rojas[2], Daniela Calderaro[2], Pai Ching Yu[2], Ivan Benaduce Casella[4], Nelson de Luccia[4], Bruno Caramelli[2]*

1 Department of Medicine, ABC Medical College University Center, Santo André, São Paulo, Brazil,
2 Unidade de Medicina Interdisciplinar em Cardiologia, Instituto do Coração, Hospital das Clínicas HCFMUSP, Faculdade de Medicina, Universidade de Sao Paulo, Sao Paulo, São Paulo, Brasil,
3 Department of Cardiology and Cardiovascular Research Institute Basel (CRIB), University Hospital Basel, Universitätsspital CH, Basel, Switzerland, 4 Vascular and Endovascular Surgery Division, Clinics Hospital, University of São Paulo Medical School, São Paulo, São Paulo, Brazil

☯ These authors contributed equally to this work.
* bcaramel@usp.br

**Data Availability Statement:** All relevant data are within the paper and its Supporting information files.

## Abstract

### Background

Conflicting results are reported about daytime variation on mortality and cardiac outcomes after non-cardiac surgeries. In this cohort study, we evaluate whether the period of the day in which surgeries are performed may influence all-cause mortality and cardiovascular outcomes in patients undergoing non-cardiac arterial vascular procedures.

### Methods

1,267 patients who underwent non-cardiac arterial vascular surgeries between 2012 and 2018 were prospectively included in our cohort and categorized into two groups: morning (7 a.m. to 12 a.m., 79%) and afternoon/night (12:01 p.m. to 6:59 a.m. in the next day, 21%) surgeries. Primary endpoint was all-cause mortality within 30 days and one year. Secondary endpoints were the incidence of perioperative myocardial injury/infarction (PMI), and the incidence of major adverse cardiac events (MACE, including acute myocardial infarction, acute heart failure, arrhythmias, cardiovascular death) at hospital discharge.

### Results

After adjusting for confounders in the multivariable Cox proportional regression, all-cause mortality rates at 30 days and one year were higher among those who underwent surgery in the afternoon/night (aHR 1.6 [95%CI 1.1–2.3], P = 0.015 and aHR 1.7 [95%CI 1.3–2.2], P < 0.001, respectively). Afternoon/night patients had higher incidence of PMI (aHR 1.4 [95%CI 1.1–1.7], P < 0.001). There was no significant difference in the incidence of MACE (aHR 1.3 [95%CI 0.9–1.7], P = 0.074).

**Funding:** TA received grant from Fundação de Amparo à Pesquisa do Estado de São Paulo (FAPESP, São Paulo, Brazil) [grant number 2019/17036-4]. URL: https://fapesp.br/ The funder had no role in the study design, data collection and analysis, decision to publish, or preparation of the manuscript.

**Competing interests:** We have read the journal's policy and the authors of this manuscript have the following competing interests: Thiago Artioli received grant from Fundação de Amparo à Pesquisa do Estado de São Paulo (FAPESP) [grant number 2019/17036-4]. Dr. Gualandro received grant from FAPESP and consulting fees from Roche, outside the submitted work. Dr. Francisco Akira Malta Cardozo received honoraria for lectures from Bayer. Dr. Calderaro received payment for lectures from Bayer, Daiichi Sankyo and Janssen; support for attending international meetings: Bayer and Daiichi Sankyo; participation on data safety monitoring board: Bayer. Dr. Ivan Benaduce Casella received consulting fees and honoraria for lectures from Bayer, Daiichi Sankyo and Boehringer Ingelheim. Dr. Bruno Caramelli received an unrestricted grant from the National Council for Scientific and Technological Development – CNPQ [grant number 304352/2016-0]. This does not alter our adherence to PLOS ONE policies on sharing data and materials.

## Conclusions

In patients undergoing arterial vascular surgery, being operated in the afternoon/night was independently associated with increased all-cause mortality rates and incidence of perioperative myocardial injury/infarction.

## Introduction

Cardiovascular disease is a major postoperative cause of morbidity and mortality in patients submitted to arterial vascular surgery.[1] The high prevalence of atherosclerosis and the coexistence of several cardiovascular risk factors increase mortality and postoperative cardiac complications [1, 2].

In the perioperative period, acute myocardial infarction (AMI), acute heart failure, cardiac arrhythmias and perioperative myocardial injury/infarction (PMI) are associated with increased length of hospital stay, elevated costs and increased mortality [3, 4]. PMI was recently described as an important complication after non-cardiac surgeries, being independently associated with high mortality rates in short and long follow-up [5–8]. Developing strategies to reduce the occurrence of perioperative cardiac complications and to better understand PMI is essential to improve outcomes for this group of high risk patients.

Overall, cardiovascular events can be influenced by the circadian cycle, with higher rates and worse outcomes of events occurring in the early morning [9–13]. However, there is still a gap in the literature assessing daytime variation of cardiovascular outcomes in the perioperative period. The variations on blood pressure, heart rate, hormone levels and many other pathophysiological mechanisms might affect perioperative outcomes [14]. A recent pilot study including patients submitted to aortic valve replacement assessed whether this time pattern occur after cardiac surgeries, showing that the incidence of major cardiovascular complications in patients undergoing surgery in the afternoon is lower [15]. Likewise, a cohort study conducted to assess daytime variation in cardiovascular events after non-cardiac surgeries did not find an influence of the time of the surgery on PMI, but the incidence of acute myocardial infarction was higher in the afternoon group [16].

On the other hand, several studies have shown worse outcomes in surgeries performed at weekends or holidays even after correcting for surgeon experience. The smaller levels of staff, consulting services and diagnostics exams available at night or weekends might result in more surgery complications [17].

In non-cardiac arterial vascular surgeries, there is no conclusive data about the association of cardiovascular complications and the period of the day in operating room. Our goal is to evaluate whether the period of the day in which surgeries are performed may influence all-cause mortality and major cardiac complications in patients undergoing non-cardiac arterial vascular interventions.

## Materials and methods

### Study overview and population

This is part of a prospective cohort registry including consecutive patients undergoing non-cardiac arterial vascular surgery between 2012 and 2018 at *Hospital das Clínicas*, University of São Paulo Medical School, São Paulo, Brazil, for whom a preoperative cardiologic consultation was requested. Cardiology consultation and perioperative evaluation is provided in less than

24 hours in our hospital. For this study, we excluded patients from whom it was not possible to perform perioperative cardiovascular evaluation or to obtain time of the surgery with the available documentation. We included patients undergoing all arterial vascular surgeries: open or endovascular (for aorta, peripheral artery, visceral artery, and carotid artery diseases and amputations due to limb ischemia), emergent, urgent, or elective. Of note, in the present study, urgent/emergency procedures were considered interventions for acute onset pathologies or clinical deterioration without previously schedule or outpatient evaluation that were able to receive perioperative cardiac evaluation.

## Procedures

Baseline characteristics of the patients were obtained from the clinical and laboratorial records registered in the preoperative cardiologic consultation. Perioperative surveillance with high-sensitive cardiac Troponin T (hs-cTnT) was performed, according to current guidelines [18, 19]. Perioperative surveillance included serial measurements of hs-cTnT (Roche Diagnostics, Indianapolis, Ind) once daily up to the second or third day after surgery and 12-lead electrocardiography daily. Additional electrocardiography and hs-cTnT measurements were performed whenever clinically indicated. The patient was included in the register just once in the index operations. Reoperations in the same hospitalization were not included.

Variables of interest from all screened patients are registered in a dedicated prospective database. Since the Revised Cardiac Risk Index already contains heart failure as one of its variables, chronic heart failure was not included in the multivariable analysis.

The exact time of surgery was obtained from the anesthetic record. In cases of impossibility to obtain the anesthetic record, the exact time of the surgery was obtained from the records of the nursing staff of the operating room (time of entry into the operating room).

## Ethical approval

This study was performed in line with the principles of the Declaration of Helsinki. Study protocol was approved by the Local Research Ethics Committee of the *Hospital das Clínicas*, University of São Paulo Medical School, Brazil (approval number: 30217414.5.0000.00068). The need for Informed Consent was waived by the ethics committee.

We adhered to the STROBE guidelines for observational studies (S1 Table).

## Definition of "morning" and "afternoon/night" surgeries

Before the analysis, we defined surgeries in the "morning" period as procedures for which the first incision occurred between 7 a.m. and 12 a.m.; and surgeries in the "afternoon/night" period as procedures for which the first incision occurred between 12:01 p.m. and 6:59 a.m. of the next day. These time intervals reflect the Hospital work shifts of the surgical at our study center and are an example of the pattern of categorization between morning and afternoon surgeries previously described in the literature [15, 16].

## Definition of study endpoints

Primary endpoint was all-cause mortality within 30 days and one year follow-up. 30-days follow-up was made by in-hospital stay registries, postoperative consultation data or telephone calls. One year follow-up for all-cause mortality was made by telephone calls, patient's charts and local death registries (with the appropriate death date confirmed).

Secondary endpoints included the incidence of perioperative myocardial injury/infarction (PMI) and the incidence of major adverse cardiac events (MACE). PMI and MACE were followed-up until hospital discharge by in-hospital stay registries.

PMI was defined as an absolute delta of hs-cTnT concentrations $\geq$ 14ng/L above the baseline concentrations within 3 days after the operation. In the absence of a preoperative hs-cTnT, PMI was defined as a delta $\geq$ 14ng/L between two postoperative concentrations [6].

MACE was a composite endpoint, including acute myocardial infarction, acute heart failure, new or decompensated arrhythmias requiring treatment, and cardiovascular death within 30 days after the surgery, which were defined according to the following criteria:

1. Acute myocardial infarction (AMI): defined according to the 4th universal definition of AMI [18]. AMI is diagnosed in the presence of elevation and decrease of myocardial necrosis markers (troponin) above reference value associated with at least one of the following criteria: I. Symptoms of myocardial ischemia; II. Electrocardiographic signs compatible with ischemia; III. New segmental change on the echocardiogram or evidence of myocardial injury in imaging studies; IV. Coronary angiography with acute coronary lesion. All AMI cases were adjudicated by two independent cardiologists.

2. Acute Heart Failure: diagnosed by cardiologist or attending physician, using clinical symptoms, physical examination findings, chest radiography, BNP or serum NT-proBNP and echocardiography, according to current guidelines [20, 21].

3. Arrhythmias (atrial fibrillation/flutter, supraventricular tachycardia, ventricular tachycardia): diagnosed when considered clinically significant, i.e., drug therapy or electrical cardioversion.

4. Cardiovascular death: deaths were classified as cardiovascular or non-cardiovascular according to guidelines. Deaths were considered cardiovascular, unless evidence of a non-cardiovascular cause was documented [22].

## Statistical analysis

Categorical variables were presented by numbers and percentages (frequencies) and were compared using Pearson's Chi-squared test with Yates' continuity. Continuous variables were presented as medians and interquartile ranges and were compared using the Mann-Whitney test (tabular results, unpaired), assuming non normal distribution.

All-cause mortality was described between patients operated in the morning vs in the afternoon/night in a descriptive analysis, with P-values calculated by Log-Rank test, and Hazard-Ratios (HR) with 95% Confidence Intervals (CI) calculated by Univariate Cox regression analysis. Both 30 days and one year follow-up were considered in the analysis.

PMI was described between patients operated in the morning vs in the afternoon/night in a descriptive analysis, with P-values calculated by Pearson's Chi-squared test with Yates' continuity (x2 test), and Odds-Ratio (OR) with 95% CI calculated by Baptista-Pike method. MACE were described between patients operated in the morning vs in the afternoon/night in a descriptive analysis, with P-values calculated by Log-Rank test, and Hazard-Ratios (HR) with 95% Confidence Intervals (CI) calculated by Univariate Cox regression analysis.

For the multivariable analysis, we calculated the adjusted Hazard-Ratios (aHR) and 95% CI using Multivariate Cox regression analysis to adjust for confounding variables. Our Regression Model and population was considered strong enough to estimate the effect size of each variable/confounder, allowing to identify the interaction effects between treatment and confounders. Significant clinical and laboratory variables in the univariable analysis

were included in the multivariable model. Based on the number of events and the consensus of requiring 10 events for each independent variable, we could address all variables with a P < 0.05 in the univariable analysis. Loss of follow-up and missing data are indicated in the descriptive text before tables and figures. No imputation was performed for missing values.

The statistical significance level considered was 95% (P < 0.05). Statistical analyses were performed using the software R for Windows, R Core Team (2020—R: A language and environment for statistical computing. R Foundation for Statistical Computing, Vienna, Austria. URL https://www.R-project.org/).

## Sensitivity analysis

Sensitivity Analysis—Since urgent/emergency surgeries are more frequent in the night period they could be correlated variables. To avoid confounding results a sensitivity analysis was made considering only the population undergoing urgent/emergency surgery, submitting them to the same statistical analysis performed in the main cohort.

## Results

### Study cohort

Between 2012 and 2018, 1,296 patients went to preoperative cardiologic consultation before arterial vascular surgery (S1 Fig), of which 29 patients were excluded, leaving 1,267 for inclusion in our cohort (S2 Fig).

### Baseline characteristics

Of the 1,267 patients, 1,002 (79.1%) underwent arterial vascular surgery in the morning and 265 (20.9%) in the afternoon/night period (Table 1). Urgent/emergency surgeries were more prevalent in the afternoon/night group. Male gender patients were more prevalent in both groups with a median age of 68 years. Patients in the afternoon/night group had more comorbidities such as diabetes mellitus requiring insulin, hypertension, chronic heart failure, had more preoperative use of clopidogrel, and less use of aspirin. Finally, the morning group had higher hemoglobin levels.

### Primary endpoint: All-cause mortality

Of 1,267 patients included in the cohort, one year follow-up was complete in 91,9%. Overall, 152 (12.0%) patients died after 30 days and 260 (20.5%) after one-year (Table 2). Significant increase in all-cause mortality in the afternoon/night surgery group was observed after 30 days (HR 2.29 [95%CI 1.65–3.19], P-value < 0.001) and one year (HR 2.38 [95%CI 1.85–3.07], P-value < 0.001).

After adjusting for confounding variables in the multivariable analysis, all-cause mortality at 30 days remained higher among those who underwent surgery in the afternoon/night period (aHR 1.6 [95%CI 1.1–2.3], P = 0.015; Table 3). The increased mortality persisted after one year in those that had surgery in the afternoon/night (aHR 1.7 [95%CI 1.3–2.2], P < 0.001; Table 3). Mortality curves are depicted in Fig 1.

Patients with higher Revised Cardiac Risk Index classifications had higher incidence of all-cause mortality. On the other hand, higher hemoglobin levels were associated with decreased incidences of mortality in both 30 days and one year.

**Table 1. Baseline characteristics of the 1,267 patients in the cohort.**

| | All Patients | Afternoon surgery | Morning surgery | P-value |
|---|---|---|---|---|
| | **n = 1267** | **n = 265** | **n = 1002** | |
| | **n (%)** | **n (%)** | **n (%)** | |
| **Male gender**, n (%) | 881 (69.5%) | 176 (66.4%) | 705 (70.4%) | 0.244 |
| **Age** (years), median (IQR) | 68 [61–75] | 68 [61.5–75] | 68 [61–75] | 0.937 |
| **Diabetes mellitus**, n (%) | | | | |
| No insulin, n (%) | 330 (26.0%) | 80 (30.2%) | 250 (25.0%) | 0.068 |
| Insulin, n (%) | 160 (12.6%) | 52 (19.6%) | 108 (10.8%) | |
| **Smoking**, n (%) | | | | |
| Smoker, n (%) | 265 (20.9%) | 54 (20.4%) | 211 (21.1%) | 0.587 |
| Former smoker, n (%) | 668 (52.7%) | 124 (46.8%) | 544 (54.3%) | |
| **Hypertension**, n (%) | 1079 (85.2%) | 235 (88.7%) | 844 (84.2%) | 0.087 |
| **Coronary artery disease**, n (%) | 482 (38.0%) | 103 (38.9%) | 379 (37.8%) | 0.810 |
| **COPD**, n (%) | 78 (6.2%) | 17 (6.4%) | 61 (6.1%) | 0.957 |
| **Chronic heart failure**, n (%) | 219 (17.3%) | 60 (22.6%) | 159 (15.9%) | **0.013** |
| **Stroke/transient ischemic attack**, n (%) | | | | |
| TIA, n (%) | 52 (4.1%) | 8 (3.0%) | 44 (4.4%) | 0.130 |
| Stroke, n (%) | 265 (20.9%) | 70 (26.4%) | 195 (19.5%) | |
| **Urgent/Emergency Surgery**, n (%) | 478 (37.7%) | 181 (68.3%) | 297 (29.6%) | **< 0.001** |
| **Revised cardiac risk index** | | | | |
| I, n (%) | 94 (7.4%) | 32 (12.1%) | 62 (6.2%) | **0.006** |
| II, n (%) | 480 (37.9%) | 90 (34.0%) | 390 (38.9%) | |
| III, n (%) | 391 (30.9%) | 75 (28.3%) | 316 (31.5%) | |
| IV, n (%) | 302 (23.8%) | 68 (25.7%) | 234 (23.4%) | |
| **Preoperative Medications** | | | | |
| ASA, n (%) | 1014 (80.0%) | 200 (75.5%) | 814 (81.2%) | 0.093 |
| Clopidogrel, n (%) | 93 (7.3%) | 30 (11.3%) | 63 (6.3%) | **0.006** |
| Statins, n (%) | 1164 (91.9%) | 219 (82.6%) | 945 (94.3%) | **< 0.001** |
| ACEI, n (%) | 401 (31.6%) | 82 (30.9%) | 319 (31.8%) | 0.962 |
| Angiotensin receptor blockers, n (%) | 293 (23.1%) | 57 (21.5%) | 236 (23.6%) | 0.619 |
| **Betablockers, n (%)** | | | | |
| Started before surgery, n (%) | 50 (3.9%) | 7 (2.6%) | 43 (4.3%) | 0.392 |
| Chronic, n (%) | 576 (45.5%) | 111 (41.9%) | 465 (46.4%) | |
| Withdraw before surgery, n (%) | 26 (2.1%) | 7 (2.6%) | 19 (1.9%) | |
| **Laboratory assessment** | | | | |
| Creatinine (mg/dL), median [IQR] | 1.11 [0.88–1.42] | 1.08 [0.84–1.46] | 1.12 [0.89–1.41] | 0.403 |
| Hemoglobin (g/dL), median [IQR] | 12.80 [11.10–14.20] | 12.20 [9.93–13.80] | 13.00 [11.40–14.30] | **< 0.001** |

IQR: interquartile range; COPD: chronic obstructive pulmonary disease; TIA: transient ischemic attack; ASA: acetylsalicylic acid; ACEI: Angiotensin-converting enzyme inhibitors.

## Secondary endpoints: PMI and MACE incidences

For this analysis, 88 (6.9%) patients did not have two consecutive values of hs-cTnT for the evaluation of PMI and were considered missing data. Overall, 351 (27.7%) patients had PMI after surgery. The incidence of PMI was higher in patients of the afternoon/night group than in patients of the morning group: 37.4% versus 25.1%, respectively (OR 1.77 [95%CI 1.34–2.36], P-value < 0.001).

**Table 2. All-cause mortality after 30 and one year of follow-up and MACE.**

|  | All Patients | Afternoon surgery | Morning surgery | HR (95%CI) | P-value |
|---|---|---|---|---|---|
|  | n = 1267 | n = 265 | n = 1002 |  |  |
|  | n (%) | n (%) | n (%) |  |  |
| **All-cause mortality 30 days** | **152 (12.0%)** | **55 (20.8%)** | **97 (9.7%)** | **2.29 [1.65–3.19]** | **< 0.001** |
| **MACE Hospital Discharge** | **302 (23.8%)** | **97 (36.6%)** | **205 (20.5%)** | **1.93 [1.52–2.46]** | **< 0.001** |
| Cardiovascular Death | 73 (5.8%) | 20 (7.5%) | 53 (5.3%) | 1.42 [0.85–2.38] | 0.200 |
| Myocardial infarction | 147 (11.6%) | 48 (18.1%) | 99 (9.9%) | 1.87 [1.32–2.64] | < 0.001 |
| Acute heart failure | 132 (10.4%) | 59 (22.3%) | 73 (7.3%) | 3.32 [2.35–4.68] | < 0.001 |
| Arrhythmia | 72 (5.7%) | 24 (9.1%) | 48 (4.8%) | 1.94 [1.19–3.17] | 0.007 |
|  | **All Patients** | **Afternoon surgery** | **Morning surgery** | **HR (95%CI)** | **P-value** |
|  | n = 1164 | n = 238 | n = 926 |  |  |
|  | n (%) | n (%) | n (%) |  |  |
| **All-cause mortality one year** | **260 (20.5%)** | **92 (34.7%)** | **168 (16.8%)** | **2.38 [1.85–3.07]** | **< 0.001** |

MACE: major adverse cardiac events; HR: hazard ratio.

**Table 3. Multivariable analysis by Cox regression model—All-cause mortality.**

|  | Adjusted HR (95%CI) 30 days | P-value | Adjusted HR (95%CI) One year | P-value |
|---|---|---|---|---|
| **All-cause mortality** |  |  |  |  |
| **Afternoon surgery** | **1.58 (1.09–2.29)** | **0.015** | **1.68 (1.27–2.22)** | **< 0.001** |
| Urgent/Emergency Surgery | **1.51 (1.02–2.23)** | **0.041** | 1.34 (0.99–1.81) | 0.055 |
| Revised Cardiac Risk Index I | - | - | - | - |
| Revised Cardiac Risk Index II | 1.54 (0.80–2.98) | 0.196 | 1.22 (0.77–1.95) | 0.395 |
| Revised Cardiac Risk Index III | 1.72 (0.87–3.39) | 0.118 | 1.28 (0.79–2.08) | 0.323 |
| Revised Cardiac Risk Index IV | 1.81 (0.92–3.58) | 0.086 | **1.67 (1.04–2.70)** | **0.035** |
| Clopidogrel | 0.59 (0.28–1.23) | 0.157 | 0.84 (0.52–1.37) | 0.489 |
| Statins | 0.64 (0.39–1.05) | 0.077 | 0.69 (0.46–1.03) | 0.071 |
| Hemoglobin (g/dL) | **0.85 (0.78–0.92)** | **< 0.001** | **0.81 (0.76–0.86)** | **< 0.001** |

aHR: adjusted hazard ratio.

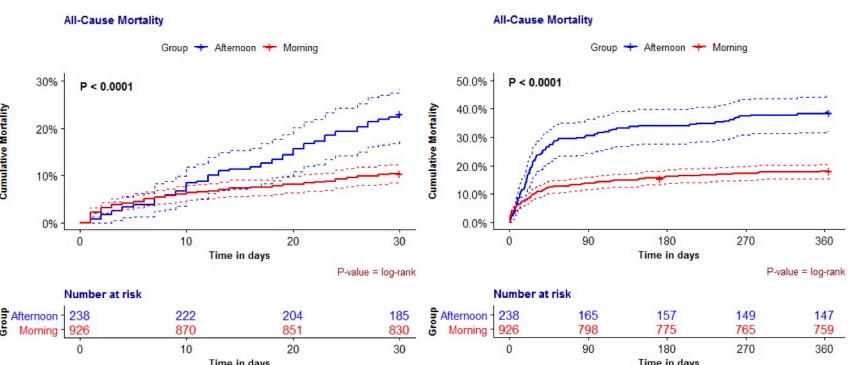

**Fig 1. All-cause mortality after arterial vascular surgery.** Groups according to the start time of the procedure in the cohort population in 30 days (A) and one-year (B) follow-up.

**Table 4. Multivariable analysis by Cox regression model—PMI and MACE.**

| | Adjusted HR | P-value |
|---|---|---|
| | (95%CI) | |
| **PMI** | | |
| **Afternoon surgery** | **1.35 (1.05–1.74)** | **0.020** |
| Urgent/Emergency Surgery | **1.78 (1.39–2.28)** | **< 0.001** |
| Revised Cardiac Risk Index I | - | - |
| Revised Cardiac Risk Index II | **2.23 (1.30–3.84)** | **0.004** |
| Revised Cardiac Risk Index III | **2.68 (1.55–4.65)** | **< 0.001** |
| Revised Cardiac Risk Index IV | **3.14 (1.81–5.44)** | **< 0.001** |
| Clopidogrel | **0.35 (0.20–0.62)** | **< 0.001** |
| Statins | **0.64 (0.44–0.93)** | **0.021** |
| Hemoglobin (g/dL) | **0.93 (0.88–0.97)** | **0.003** |
| **MACE** | | |
| **Afternoon surgery** | 1.27 (0.98–1.66) | 0.074 |
| Urgent/Emergency Surgery | **1.71 (1.30–2.26)** | **< 0.001** |
| Revised Cardiac Risk Index I | - | - |
| Revised Cardiac Risk Index II | 1.39 (0.89–2.18) | 0.152 |
| Revised Cardiac Risk Index III | 1.45 (0.91–2.31) | 0.117 |
| Revised Cardiac Risk Index IV | **1.66 (1.04–2.65)** | **0.034** |
| Clopidogrel | 0.73 (0.46–1.17) | 0.193 |
| Statins | **0.44 (0.31–0.61)** | **< 0.001** |
| Hemoglobin (g/dL) | **0.85 (0.80–0.90)** | **< 0.001** |

aHR: adjusted hazard ratio. PMI: perioperative myocardial injury/infarction. MACE: major adverse cardiac events.

In the multivariable analysis, surgeries that occurred in the afternoon/night group were also independently associated with higher PMI incidence (Table 4). Urgent/emergency surgeries were associated with higher PMI rates. Also, patients with higher Revised Cardiac Risk Index classifications had higher incidence of PMI. Preoperative clopidogrel was associated with lower incidence of PMI.

At hospital discharge, the incidence MACE was higher in the afternoon/night surgery group (Table 2). Surgeries occurring in the afternoon/night period were not independently associated with increased MACE after correction for confounders (Table 4). However, urgent/emergency surgeries and higher Revised Cardiac Risk Index classifications were associated with elevated occurrence of MACE. The use of preoperative statins and higher hemoglobin levels were associated with lower rates of MACE.

## Sensitivity analysis

Of the total, 478 patients underwent urgent/emergency surgery: 181 (37.9%) during the morning and 297 (62.1%) during the afternoon/night period. Considering only urgent/emergency surgery population, significant increase in all-cause mortality in the afternoon/evening surgery group was observed after 30 days (HR 2.57 [95% CI 1.69–3.93], P value < 0.001) and one year (HR 2.43 [95% CI 1.75–3.37], P value < 0.001). After multivariate analysis, 30-day all-cause mortality was higher in the afternoon/night period (aHR 2.43 [95%CI 1.54–3.83], P = < 0.001), as well as one year mortality (aHR 2.35 [95%CI 1.65–3.33], P = < 0.001)–(S3 Fig).

The incidence of PMI in the urgent/emergency surgery group was also higher in patients operated in the afternoon/evening period: 44.2% versus 33.0%, respectively (OR 1.82 [95%CI

1.22–2.71], P- value = 0.003). In the multivariate analysis, urgent/emergency surgeries performed in the afternoon/evening presented independent association with the incidence of PMI (aHR 1.4 [95%CI 1.03–1.91], P = 0.03). After 30 days, the incidence of MACE in urgent/emergency surgery group was higher in the afternoon/evening surgery population. Surgeries performed in the afternoon/evening period were independently associated with increased MACE after correction for confounding factors in the multivariable analysis (aHR 1.51 [95% CI 1.10–2.06], P = 0.010).

## Discussion

In this prospective cohort study, we aimed to investigate whether the period of the day in which surgeries are performed may influence all-cause mortality, perioperative myocardial injury/infarction (PMI) and major adverse cardiac events (MACE) in patients undergoing non-cardiac arterial vascular procedures. We observed that surgeries performed in the afternoon/night period were associated with higher all-cause mortality at 30 days and one year and elevated PMI incidence (Fig 2). The incidence of MACE at hospital discharge, however, was not different between morning and afternoon/night surgical groups.

The higher mortality rates in the afternoon/night group during follow-up presented in our study was not previously described. While a recent study by du Fay de Lavallaz *et al* [16] found a higher acute myocardial infarction (AMI) incidence in the afternoon group and no difference in mortality according to the time of surgery, our study observed an almost 18% higher mortality rate in those operated in the afternoon / night without any difference in MACE incidence. We did not evaluate the incidence of MACE at long term follow-up, thus we could have missed some events. Also, since we had a higher-risk population with patients undergoing vascular surgeries that need cardiac referral, the elevated all-cause mortality rates observed may have overshadow other cardiac events. These results differs from the findings of a recent pilot study involving daytime variation on cardiac surgeries (on-pump

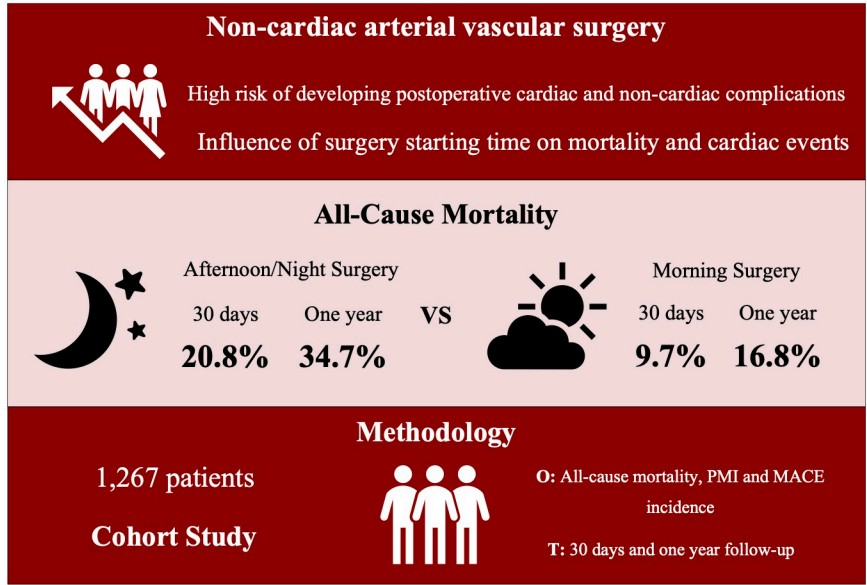

**Fig 2. Graphical abstract.** In patients submitted to arterial vascular surgery, being operated in the afternoon/night period was independently associated with increased mortality rates and higher incidence of PMI. O: outcomes. T: time of follow-up.

aortic valve replacement), which demonstrated higher postoperative morbidity in patients operated in the morning [15].

The disparities between the studies might be explained because of different patient populations, human factors, and structural procedures. Compared with the other two cohorts, our patients had more comorbidity such as diabetes, hypertension, coronary artery disease, chronic heart failure and cerebrovascular arterial disease. Regarding the type of the surgery, our study focused on a specific type of higher risk surgery: arterial vascular surgeries. Also, urgent/emergency procedures were more prevalent than in the other studies. We decided to include urgent/emergency surgeries in our cohort because they represent a significant share of all vascular surgeries performed in our service. Therefore, we aimed to evaluate the same endpoints of elective surgeries to have a perception of how they would perform. Although the proportion of urgent/emergency surgeries could weight in the Cox Regression, we considered that our Regression Model and population was strong enough to estimate the effect size of each variable/confounder. At last, our study population is based on a tertiary health care center to which patients are usually referred when they have a more severe vascular condition or important cardiac or noncardiac comorbidities, often arriving at a poor clinical condition. Those combined situations could explain the high all-cause mortality rate overcoming the impact of MACE in the perioperative period.

Despite our standardized perioperative care management, it is possible that other non-measured factors might have influenced the outcomes, such as staffing, care units and other human factors. Not only patients' characteristics and clinical aspects are responsible for time distribution; operating issues and surgeries delay could affect the schedule of the surgeries, which were not assessed in our study. A study by van Zaane *et al* could not identify association between in-hospital mortality and the time of the day in urgent non-cardiac surgeries [23], but still we have to consider that the fatigue level of the surgeons and staff was not measured in our cohort, and this could be an important factor to explain our worse afternoon prognosis, since the "afternoon/night" period included hours from 12:01 p.m. and 6:59 a.m. Moreover, care units may have fewer employees working on afternoon/night shifts and, also, those working on late shifts could be less experienced. In fact, several studies have already demonstrated worse surgery outcomes in patients operated in weekends or holidays [24, 25]. Finally, for elective surgeries, our hospital have the protocol of 8 hours of fasting before the procedure. However, due to delays and other non-measured factors patients might have been exposed to longer periods of fasting, especially for afternoon/night procedures. Prolonged fasting might have contributed to the observed outcomes [26, 27]. The fasting period of urgent/emergency surgeries was not measured.

Our finding of daytime variation on PMI incidence is also different from the reported on previous cohorts in non-cardiac surgery [16, 28], which found no difference in the incidence of PMI when comparing morning vs afternoon. These studies on cardiovascular function, biorhythms and cardiac disorders still have many uncertainties in their conclusions. One plausible explanation is that the pathophysiological mechanisms involved in PMI are different depending on the type of procedure. Further studies are necessary to better understand the mechanisms related to perioperative myocardial injury.

Other secondary finding of our study was the association between preoperative use of statins and better prognosis of MACE at hospital discharge. Our cohort corroborate data from previous studies in which statins demonstrated to be effective in the prevention of cardiovascular events after vascular surgeries [29]. Additionally, anemia was usually associated with worse outcomes in terms of mortality and MACE, another well-known marker of frailty and worse perioperative outcomes [30].

Our study has some limitations. First, our study population involves patients from a tertiary health care center to which patients are usually referred when they have a more severe vascular condition or important cardiac or noncardiac comorbidities. We included in the cohort patients for whom cardiac consultation was requested, and although it is a common practice at our institution for vascular surgeons to request cardiac evaluation for all patients, we may have missed patients in the lower risk strata. In clinical practice, low risk patients according to anesthesiology evaluation are often directly sent to surgical theater without further consultation. Given the nature of the present study (retrospective analysis of prospective collected data) we don't have the number of patients operated without cardiac consultation, but we estimate a small number of cases that could not influence the final results of the present study. Second, despite having used a regression model to adjust for confounders, we must consider that other unmeasured confounders may have interacted with the results. Third, our study was made based on a specific type of surgery: arterial vascular surgeries. Although this enhances the power of the study and avoid possible confounders, such as different perioperative risks, our findings cannot be extrapolated for other types of surgery. Fourth, although we have considered urgent/emergency surgeries in our analysis, we have not measured the level of stress or tiredness and the level of experience of the healthcare professionals who performed the surgeries, which may add some degree of bias in our study.

The association between mortality rates, PMI, MACE and the period of the day in which surgeries are performed seem to be a promising field of study. The results of this cohort may lead to readjustments of simple medical procedures (such as determining the best time for the surgical procedure, which may indicate better prognosis) or in monitoring the patient (for example, determining how the multidisciplinary follow-up of a high-risk cardiovascular patient undergoing vascular surgery should be), simple measures that may have an impact on improving clinical practice. The cardioprotective strategy based only on the time of surgery indicates an economical and easy to implement measure for the prevention of postoperative cardiovascular complications.

## Conclusion

In high-risk patients submitted to arterial vascular surgeries, being operated in the afternoon/night period was independently associated with increased all-cause mortality rates and higher incidence of PMI. We could not identify any influence of the period of the day in which surgeries are performed on the incidence of MACE.

## Supporting information

**S1 Fig. Distribution by surgery type from morning and afternoon/night groups.**
AAAE = Abdominal Aortic Aneurysm Endovascular; AAAO/H = Open or Hybrid Abdominal Aortic Aneurysm; TAAEO/H = Thoracic Aortic Aneurysm Endovascular, Open or Hybrid; LLRO/E = Open or Endovascular Lower Limb Revascularization; EC = Endovascular Carotid; OC = Open Carotid; AMPUT = Amputations; OTHERS = Open or Endovascular Visceral Arteries and Other Procedures.
(TIF)

**S2 Fig. Study population.**
(TIF)

**S3 Fig. Mortality in urgent/emergency surgeries.**
(TIF)

**S1 Table. STROBE statement—Checklist of items that should be included in reports of cohort studies.**
(DOCX)

## Author Contributions

**Conceptualization:** Thiago Artioli, Danielle Menosi Gualandro, Bruno Caramelli.

**Data curation:** Thiago Artioli, Francisco Akira Malta Cardozo, María Carmen Escalante Rojas.

**Formal analysis:** Thiago Artioli, Danielle Menosi Gualandro, María Carmen Escalante Rojas, Bruno Caramelli.

**Funding acquisition:** Nelson de Luccia, Bruno Caramelli.

**Investigation:** Thiago Artioli.

**Methodology:** Danielle Menosi Gualandro, María Carmen Escalante Rojas, Ivan Benaduce Casella, Bruno Caramelli.

**Project administration:** Danielle Menosi Gualandro, Ivan Benaduce Casella, Nelson de Luccia, Bruno Caramelli.

**Resources:** Francisco Akira Malta Cardozo, Ivan Benaduce Casella, Nelson de Luccia, Bruno Caramelli.

**Software:** María Carmen Escalante Rojas.

**Supervision:** Danielle Menosi Gualandro, Francisco Akira Malta Cardozo, Daniela Calderaro, Pai Ching Yu, Ivan Benaduce Casella, Bruno Caramelli.

**Validation:** Francisco Akira Malta Cardozo, Daniela Calderaro, Pai Ching Yu, Ivan Benaduce Casella, Nelson de Luccia.

**Visualization:** Francisco Akira Malta Cardozo, Daniela Calderaro, Pai Ching Yu, Ivan Benaduce Casella.

**Writing – original draft:** Thiago Artioli, Danielle Menosi Gualandro, María Carmen Escalante Rojas, Ivan Benaduce Casella, Nelson de Luccia.

**Writing – review & editing:** Thiago Artioli, Danielle Menosi Gualandro, Nelson de Luccia, Bruno Caramelli.

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
