## [Decision Letter · Decision Letter 0]

23 Sep 2021

PONE-D-21-21807Impact of the period of the day on all-cause mortality and major cardiovascular complications after arterial vascular surgeriesPLOS ONE

Dear Dr. Caramelli,

Thank you for submitting your manuscript to PLOS ONE. After careful consideration, we feel that it has merit but does not fully meet PLOS ONE’s publication criteria as it currently stands. Therefore, we invite you to submit a revised version of the manuscript that addresses the points raised during the review process.

Please, carefully address all comments and suggestions  and return a revised  version at your earlier convenience.

We look forward to receiving your revised manuscript.

Kind regards,

Cesario Bianchi

Academic Editor

PLOS ONE

Journal Requirements:

2. Please provide additional details regarding participant consent. In the Methods section, please ensure that you have specified (1) whether consent was informed and (2) what type you obtained (for instance, written or verbal). If your study included minors, state whether you obtained consent from parents or guardians. If the need for consent was waived by the ethics committee, please include this information.

I have read the journal's policy and the authors of this manuscript have the following competing interests: Thiago Artioli received grant from Fundação de Amparo à Pesquisa do Estado de São Paulo (FAPESP) [grant number 2019/17036-4]. Dr. Gualandro received grant from FAPESP and consulting fees from Roche, outside the submitted work. Dr. Francisco Akira Malta Cardozo received honoraria for lectures from Bayer. Dr. Calderaro received payment for lectures from Bayer, Daiichi Sankyo and Janssen; support for attending international meetings: Bayer and Daiichi Sankyo; participation on data safety monitoring board: Bayer. Dr. Ivan Benaduce Casella received consulting fees and honoraria for lectures from Bayer, Daiichi Sankyo and Boehringer Ingelheim. Dr. Bruno Caramelli received an unrestricted grant from the National Council for Scientific and Technological Development – CNPQ [grant number 304352/2016-0].

Additional Editor Comments:

Dear Dr. Caramelli;

Your manuscript was reviewed by 2 experts that found it of interest. However, there are many issues from the study design and many confounding factors needed to be, seriously, addressed before we can make a final decision about your submission.

Please, carefully, answer each and every criticism and make changes to the revised version.

We look forward to the revised manuscript.

Reviewers' comments:

Reviewer's Responses to Questions

**Comments to the Author**

1. Is the manuscript technically sound, and do the data support the conclusions?

Reviewer #1: Yes

Reviewer #2: No

2. Has the statistical analysis been performed appropriately and rigorously? 

Reviewer #1: Yes

Reviewer #2: Yes

3. Have the authors made all data underlying the findings in their manuscript fully available?

Reviewer #1: Yes

Reviewer #2: Yes

4. Is the manuscript presented in an intelligible fashion and written in standard English?

Reviewer #1: Yes

Reviewer #2: Yes

5. Review Comments to the Author

Reviewer #1: Impact of the period of the day on all-cause mortality and major cardiovascular complications after arterial vascular surgeries

Artioli et al. reported data regarding the impact of the time-of-the-day on post-operative mortality and cardiovascular complications. Exploring a large cohort (>1200 patients), two subgroup were formed to categorized morning and afternoon surgeries. They observed higher rates of all-mortality at 1 month and 12 month in patient who underwent surgery in the afternoon.

The topic is of interest, because of the disparity in the results of these time-of-the-day post-operative complications studies. The discussion of the paper is well written and takes into account the major knowledge in the field.

However, the design of the study introduces some confounding factors, because the participant were not randomized.

We would suggest several points

if available data :

• Distribution with surgery schedule ? An histrogram of the distribution of surgery time-of-the-day should be provided.

• Divided Afternoon & night surgery � Recently Montaigne et al. published a paper regarding time-of-the-day and Kidney transplantation which should be discussed in line with the conclusion. It would be of interest to compare day-time and night surgery, by different medical organization ?

• Line 78 : The sentence is quite unclear. “it is poorly understood if daytime variation also occurs in the perioperative period”, please rephrase.

• Line 130 to 134 : In the paper of du Fay and collegues and Montaigne and collegues, an interval of 3 hours was observed between “morning” and “afternoon”.

• Line 154 : What is “considered clinically significant” ? High incidence of atrial fibrillation in the morning could distort the observations

• Line 202 : What type of vascular surgery ? Duration of the operation ?

• Table 1 : Please provide preoprative TnT and Left ventricular ejection fraction ?

• No exclusion criterias, why ? No flowchart ?

• Emergency surgery represent a high proportion of the afternoon group vs morning group and weights a lot in the Cox regression. Why this particular population is not excluded ? This is a major confounding factor that should be presented as a clear limitation.

• Fasting duration could be important in peri-operative studies, because of the amount of ketone bodies increasing in time. What was the fasting durations in each group (including emergency surgery) ? Was the ketonemia higher in any group ? This should be at least discussed. (Youm et al. Nat. Med. 2015)

• Is nosocomial infection an exclusion criteria from this cohort ?

• In this population (>1200 patients) case-control with propensity matching design would be of interest.

Reviewer #2: This is an interesting paper describing increased postoperative mortality both at hospital discharge and one year following vascular surgery in a tertiary centre for operations started after 12 mid-day compared to those whose operation was started in the morning between 7am and mid-day.

The paper illustrates well that those operated later in the day were sicker, higher cardiac risk and not surprisingly did less well immediately after surgery and even a year after (20.8% and 34.7% mortality !)

1. It is disappointing therefore that in the design of the study (and these results might well have been expected -what was the hypothesis?) data was not included at base line which might address the reason why operations started later in the day have sicker patients, more often are urgent/ emergent and do less well. Some factors are raised in the discussion but this study does not address the system issues.

2. The method describes that only Consecutive cases for whom a preoperative cardiological consultation was requested were included. There were 1267 patients but how many patients were there who didn’t have a cardiological consultation ? Obtaining this consultation may be affected by time of day or urgency ? Can the authors comment on this and preferably indicate how many other vascular surgical procedures were done without this consultation and what was the mortality rate for this group- to give a full picture of the context of this study.

3. In vascular surgical practice repeat operations are not uncommon how was this taken into account in the study.

4. No distinction is made regarding the nature of the vascular procedures done – it would be important to describe these as the outcomes, the open vs endovascular options, major vs minor and choice of time of day will be influence by these

5. Follow up was described both as 30 days and at -hospital discharge -which was it ? One year data was by telephone and registries. In the Survival data there are multiple steps – does this mean telephone calls were made regularly ?

6. The authors describe urgent/emergency surgery (how was this defined ? and how many of these underwent prior cardiological assessment? This was a very major difference in the two time groups 68% compared to 29.6% . The data suggests that this was the most important risk factor maybe even more so than time of day.

7. Most of the analysis is of patient characteristics and they were sicker but what of other system factors that would affect the time distribution? Was there any access to operating theatre issue – which pushed procedures in to the afternoon. Were there delays to surgery which pushed procedures to later in the day. Interestingly urgency was a major risk factor for MACE

8. The pattern of referral to this tertiary hospital was mentioned - as a possible system factor. Regional referral patterns for vascular surgery are well recognised, which may be related to availability of vascular surgery specialists and appropriate support services. If not in the data collected can the authors please comment on the factors in their hospital/region that influence what they have reported which could lead to outcome improvement.

9. Discussion page 12 line 271 it says “Additionally, our population were exclusive of patients undergoing arterial vascular surgery, which are patients at greater cardiac risk undergoing mostly high-risk operations. -could the authors please explain ? Do they mean Additionally, our population included only patients …. Or Additionally, our population were exclusively patients undergoing …..

10. The Graphical abstract fig 2 should be revisited as it misses the point of the study – This reviewer would suggest that the study is not primarily about “need for better postoperative outcomes control” (not sure what that means). The message is about operations done later in the day – How can we insure the high risk patients are better prepared and more often operated earlier in the day.

11. Similarly the Conclusion should be revisited – Vascular surgery patients operated later in the day are sicker, more urgent and have worse outcomes with double the mortality at 30m days and even out to one year. Steps are needed to improve outcomes which may include starting operations earlier.

6. PLOS authors have the option to publish the peer review history of their article (what does this mean?). If published, this will include your full peer review and any attached files.

Reviewer #1: No

Reviewer #2: No

---

## [Author Response · Author response to Decision Letter 0]

4 Nov 2021

Dear Editor and Reviewers,

We appreciate your patience and consideration on reviewing our manuscript. It is an honor for us to have the opportunity to publish in PLOS ONE. We have carefully read your comments and we changed the manuscript accordingly.

Please, note that changes we made in the manuscript text are marked as the “red” color. Also, we are providing the files of both marked and clean “Impact of the period of the day on all-cause mortality and major cardiovascular complications after arterial vascular surgeries” reviewed versions attached.

List of Changes – Journal Requirements

1. Editor: Please ensure that your manuscript meets PLOS ONE's style requirements, including those for file naming.

Authors’ Feedback: As recommended by the Editor, we have revised the requirements in our manuscript regarding style, file naming and references, and we believe it is now correct.

2. Editor: Please provide additional details regarding participant consent. In the Methods section, please ensure that you have specified (1) whether consent was informed and (2) what type you obtained (for instance, written or verbal). If your study included minors, state whether you obtained consent from parents or guardians. If the need for consent was waived by the ethics committee, please include this information.

Authors’ Feedback: The ethics committee waived the need for an informed consent in our study, because it involved analysis of medical records from the Hospital database. We have added the following sentence to the study.

Methods (page 5, line 124): “The need for Informed Consent was waived by the ethics committee.”

3. Editor: Please confirm that this does not alter your adherence to all PLOS ONE policies on sharing data and materials, by including the following statement: "This does not alter our adherence to PLOS ONE policies on sharing data and materials.” (as detailed online in our guide for authors http://journals.plos.org/plosone/s/competing-interests). If there are restrictions on sharing of data and/or materials, please state these. Please note that we cannot proceed with consideration of your article until this information has been declared. Please include your updated Competing Interests statement in your cover letter; we will change the online submission form on your behalf.

Authors’ Feedback: As recommended by the Editor, we have added the sentence "this does not alter our adherence to PLOS ONE policies on sharing data and materials” in the Competing Interests section in our cover letter.

4. Editor: Please include captions for your Supporting Information files at the end of your manuscript, and update any in-text citations to match accordingly. Please see our Supporting Information guidelines for more information: http://journals.plos.org/plosone/s/supporting-information.

Authors’ Feedback: As recommended by the Editor, we have revised the requirements in our manuscript regarding captions for Supporting Information and citations, and we believe it is now correct.

List of Changes – Reviewers’ Comments

Reviewer #1

1. Reviewer: Distribution with surgery schedule? An histogram of the distribution of surgery time-of-the-day should be provided.

Authors’ Feedback: Based on the surgical work shifts at our Hospital, predefined for the present study, we have categorized the variable time of the surgery into two groups: morning surgeries (between 7 a.m. to 12 a.m.) and afternoon/night surgeries (between 12:01p.m. to 6:59a.m. of the next day). We believe that this categorical approach might be more robust than a continuous one. Also, we believe that the categorical approach has more clinical relevance due to its feasibility and capacity to be more understandable, and it’s more related to Hospital work shifts. The histogram was added in the Suporting Material of the Manuscript.

2. Reviewer: Divided Afternoon & night surgery.

Authors’ Feedback: The decision to piece together afternoon and night surgeries was made based on our Hospital work shifts of the surgical teams. Also, night surgeries mostly included urgent/emergency procedures and it didn’t represent a significative number of surgeries so we could split and without jeopardizing the power of the study. Therefore, we made the choice to categorize it in “afternoon/night” surgeries (12:01p.m. to 6:59a.m. of the next day).

3. Reviewer: Recently Montaigne et al. published a paper regarding time-of-the-day and Kidney transplantation which should be discussed in line with the conclusion. It would be of interest to compare day-time and night surgery, by different medical organization?

Authors’ Feedback: Many thanks to the reviewer for the suggestion. Despite Montaigne’s article is related to a different surgical procedure, the idea of comparing surgical outcomes between two different period of the day is in alignment with the present study. However, as we stated in item 2, the periods of the day that were compared are different, in our study and in Montaigne’s study. As stated in item 2, in our study the choice of the period of the day to be compared were related to Hospital work shifts of the surgical team. 

4. Reviewer: Line 78: The sentence is quite unclear. “it is poorly understood if daytime variation also occurs in the perioperative period”, please rephrase.

Authors’ Feedback: We agree with the suggestion made by the reviewer. We have corrected the sentence in order to make it clearer and better understandable. The modified version is shown below:

Introduction (page 3, line 74): “However, there is still a gap in the literature assessing daytime variation of cardiovascular outcomes in the perioperative period.”

5. Reviewer: Line 130 to 134: In the paper of du Fay and colleagues and Montaigne and colleagues, an interval of 3 hours was observed between “morning” and “afternoon”.

Authors’ Feedback: We agree with the comments. To avoid confusion, we have corrected the sentence in page 5, line 128. Additionally, we decided to maintain our definition of morning surgeries (between 7 a.m. to 12 a.m.) and afternoon/night surgeries (between 12:01p.m. to 6:59a.m. of the next day) because we wanted to gather the maximum number of surgeries as possible. Considering the distribution of the time of our surgeries, very few happened in the splitting time (for example around 6a.m. or 12a.m.), and so this pattern wouldn’t underpower our categorization or results.

Methods (page 5, line 131): “These time intervals reflect the Hospital work shifts of the surgical team at our study center and are an example of the pattern of categorization between morning and afternoon surgeries previously described in the literature [15, 16].”

6. Reviewer: Line 154: What is “considered clinically significant”? High incidence of atrial fibrillation in the morning could distort the observations.

Authors’ Feedback: Clinically significant arrhythmias were defined as those in need of active drug therapy or electrical cardioversion. This definition was stablished to assess arrhythmias (one of the topics of Major Adverse Cardiac Events – MACE) which were adverse events after the arterial vascular surgeries. To make it clearer, we have added the following.

Methods (page 5, line 146): “…new or decompensated arrhythmias requiring treatment…”

7. Reviewer: Line 202: What type of vascular surgery? Duration of the operation?

Authors’ Feedback: We included patients undergoing all arterial vascular surgeries: open or endovascular (for aorta, peripheral artery, visceral artery, and carotid artery diseases and amputations due to limb ischemia), emergent, urgent, or elective. The distribution by surgery type is similar when comparing morning and afternoon, not considering urgent/emergency surgeries, which were analyzed as a confounding variable in our analysis. We have added the histogram in the Supporting Material of the Manuscript.

Results (page 8, line 220): “…arterial vascular surgery…”

8. Reviewer: Table 1: Please provide preoprative TnT and Left ventricular ejection fraction?

Authors’ Feedback: In our Hospital, preoperative Echocardiogram and cardiac troponin levels are not part of the routine exams. These exams are performed at physician’s discretion. We have checked and the minority of the patients had these exams available.

9. Reviewer: No exclusion criterias, why? No flowchart?

Authors’ Feedback: We determined that the exclusion criteria of our study is, as stated in the methodology and in the Supportive Figure 1 (our flowchart): “For this analysis, we excluded patients from whom it was not possible to obtain time of the surgery with the available documentation.” (Methods page 4, line 101) We considered that other situations, as mentioned by the reviewer in the suggestions, are contraindications for surgical procedures, not exclusion criteria.

10. Reviewer: Emergency surgery represent a high proportion of the afternoon group vs morning group and weights a lot in the Cox regression. Why is this particular population not excluded? This is a major confounding factor that should be presented as a clear limitation.

Authors’ Feedback: We thank the reviewer for the comment. However, our group decided to include urgent/emergency surgeries in our cohort because of some factors. First, urgent/emergency surgeries represent an important share of arterial vascular surgeries in our Hospital. In this way, we wanted to study the same endpoints than in elective surgeries to have a perception of how they would perform equally or differently. Second, although the proportion of urgent/emergency surgeries could weight in the Cox Regression, we considered that our Regression Model and population was strong enough to estimate the effect size of each variable/confounder, allowing to identify the interaction effects between treatment and confounders. Lastly, as recommended by the reviewer, we have added this discussion in the manuscript text.

Discussion (page 13, line 292): “Also, urgent/emergency procedures were more prevalent than in the other studies. We decided to include urgent/emergency surgeries in our cohort because they represent an important share of arterial vascular surgeries in our service. In this way, we wanted to study the same endpoints than in elective surgeries to have a perception of how they would perform. Although the proportion of urgent/emergency surgeries could weight in the Cox Regression, we considered that our Regression Model and population was strong enough to estimate the effect size of each variable/confounder.”

11. Reviewer: Fasting duration could be important in peri-operative studies, because of the amount of ketone bodies increasing in time. What was the fasting durations in each group (including emergency surgery)? Was the ketonemia higher in any group? This should be at least discussed. (Youm et al. Nat. Med. 2015)

Authors’ Feedback: It is true that fasting duration is an important factor for perioperative care. However, it is not routine in our Hospital service to previously measure ketone bodies in surgical patients. For elective surgeries, we have the protocol of 8 hours of fasting duration pre surgery. We don’t have the fasting durations in urgent/emergency surgeries. We added this to the Discussion and Limitation as described below.

Discussion (page 14, line 311): “Finally, for elective surgeries, our hospital have the protocol of 8 hours of fasting before the procedure. However, due to delays and other non-measured factors patients might have been exposed to longer periods of fasting, especially for afternoon/night procedures. Prolonged fasting might have contributed to the observed outcomes [26, 27]. The fasting period of urgent/emergency surgeries was not measured.”

12. Reviewer: Is nosocomial infection an exclusion criteria from this cohort?

Authors’ Feedback: The presence of nosocomial infection was not an exclusion criteria for our study.

13. Reviewer: In this population (>1200 patients) case-control with propensity matching design would be of interest.

Authors’ Feedback: We have considered Propensity Score Matching as a possibility to our statistical analysis as it could be an easier method to discuss with the audience, more straightforward and objective. It was done in one of the papers in our references (du Fay and colleagues). However, we considered that Propensity Score Matching wouldn’t allow us to see clear interactions between treatment and confounders, and it wouldn’t allow us to use all patients in the cohort, losing some interactions we wanted to search and losing power. Also, Propensity Score Matching would be more feasible if the number of outcome events/number of confounders were less than 7, which is not the case of our study. Therefore, we believe that Cox Regression Model is suitable to our study. We have added those points in the methods section.

Methods (page 6, line 177): “Our Regression Model and population was considered strong enough to estimate the effect size of each variable/confounder, allowing to identify the interaction effects between treatment and confounders.”

Reviewer #2

1. Reviewer: This is an interesting paper describing increased postoperative mortality both at hospital discharge and one year following vascular surgery in a tertiary centre for operations started after 12 mid-day compared to those whose operation was started in the morning between 7am and mid-day. The paper illustrates well that those operated later in the day were sicker, higher cardiac risk and not surprisingly did less well immediately after surgery and even a year after (20.8% and 34.7% mortality !). It is disappointing therefore that in the design of the study (and these results might well have been expected -what was the hypothesis?) data was not included at base line which might address the reason why operations started later in the day have sicker patients, more often are urgent/ emergent and do less well. Some factors are raised in the discussion but this study does not address the system issues.

Authors’ Feedback: We thank the reviewer for the suggestion. We have categorized the variable time of the surgery into two groups based on the surgical work shifts at our Hospital: morning surgeries (between 7 a.m. to 12 a.m.) and afternoon/night surgeries (between 12:01 p.m. to 6:59 a.m. of the next day). The null hypothesis of our study is that there is no difference in outcomes between these two groups. We modified the final version of the manuscript to make it clearer.

2. Reviewer: The method describes that only Consecutive cases for whom a preoperative cardiological consultation was requested were included. There were 1267 patients but how many patients were there who didn’t have a cardiological consultation? Obtaining this consultation may be affected by time of day or urgency? Can the authors comment on this and preferably indicate how many other vascular surgical procedures were done without this consultation and what was the mortality rate for this group- to give a full picture of the context of this study.

Authors’ Feedback: We added this information as a Limitation in the final version of the manuscript.

Discussion (page 14, line 328): “We included in the cohort patients for whom cardiac consultation was requested, and although it is a common practice at our institution for vascular surgeons to request cardiac evaluation for all patients, we may have missed patients in the lower risk strata.”

3. Reviewer: In vascular surgical practice repeat operations are not uncommon how was this taken into account in the study.

Authors’ Feedback: We added the following sentence in the methods section, so it is clearer and better understandable.

Methods (page 4, line 110): “The patient was included in the register just once in the index operations. Reoperations in the same hospitalization were not included. 

4. Reviewer: No distinction is made regarding the nature of the vascular procedures done – it would be important to describe these as the outcomes, the open vs endovascular options, major vs minor and choice of time of day will be influence by these.

Authors’ Feedback: Our group decided to categorize arterial vascular surgeries by its start time, apart from the type of the surgery. Based on the surgical service hours of our study center, we have categorized the variable time of the surgery into two groups: morning surgeries (between 7 a.m. to 12 a.m.) and afternoon/night surgeries (between 12:01p.m. to 6:59a.m. of the next day). We believe that this categorical approach, considering all arterial vascular surgeries together, might be more robust, due to its clinical relevance, feasibility and capacity to be more understandable. Although this strategy could decrease the power of our study, our statistical methods were calculated enough to diminish this situation. We have added an histogram with surgeries type in the Supporting Material of the Manuscript.

5. Reviewer: Follow up was described both as 30 days and at -hospital discharge -which was it? One year data was by telephone and registries. In the Survival data there are multiple steps – does this mean telephone calls were made regularly?

Authors’ Feedback: All-cause mortality (primary endpoint) was followed-up during 30-days after the surgery and one year after the surgery. 30-days follow-up was made by in-hospital stay registries, postoperative consultation data or telephone calls. One year follow-up for all-cause mortality was made by telephone calls, patient’s charts and local death registries (with the appropriate death date confirmed), which explain why it was possible to obtain multiple steps in the Survival analysis.

PMI and MACE (secondary outcomes) were followed-up until hospital discharge by in-hospital stay registries.

We have corrected the text in the manuscript, so it is clearer and better understandable.

Methods (page 5, line 136): “30-days follow-up was made by in-hospital stay registries, postoperative consultation data or telephone calls. One year follow-up for all-cause mortality was made by telephone calls, patient’s charts and local death registries (with the appropriate death date confirmed).”

Methods (page 5, line 141): “PMI and MACE were followed-up until hospital discharge by in-hospital stay registries.”

6. Reviewer: The authors describe urgent/emergency surgery (how was this defined? and how many of these underwent prior cardiological assessment? This was a very major difference in the two time groups 68% compared to 29.6%. The data suggests that this was the most important risk factor maybe even more so than time of day.

Authors’ Feedback: Urgent/emergency procedures were considered surgeries non-previously scheduled, which means they were not elective and not previously followed up in ambulatorial treatment. Our group decided to include urgent/emergency surgeries in our cohort because of some factors. First, urgent/emergency surgeries represent an important share of arterial vascular surgeries in our service. In this way, we wanted to study the same endpoints than in elective surgeries to have a perception of how they would perform equally or differently. Second, although the proportion of urgent/emergency surgeries could weight in the Cox Regression, we considered that our Regression Model and population was strong enough to estimate the effect size of each variable/confounder, allowing to identify the interaction effects between treatment and confounders. Urgent/emergency surgery are an important risk factor, which was something previously expected. Lastly, as recommended by the reviewer, we have added this discussion in the manuscript text.

Methods (page 4, line 99): “Urgent/emergency procedures were considered surgeries non-previously scheduled, which means they were not elective and not recently evaluated in an outcome section.”

Discussion (page 13, line 292): “Also, urgent/emergency procedures were more prevalent than in the other studies. We decided to include urgent/emergency surgeries in our cohort because they represent a significant share of all vascular surgeries performed in our service. Therefore, we aimed to evaluate the same endpoints of elective surgeries to have a perception of how they would perform. Although the proportion of urgent/emergency surgeries could weight in the Cox Regression, we considered that our Regression Model and population was strong enough to estimate the effect size of each variable/confounder.”

7. Reviewer: Most of the analysis is of patient characteristics and they were sicker but what of other system factors that would affect the time distribution? Was there any access to operating theatre issue – which pushed procedures in to the afternoon. Were there delays to surgery which pushed procedures to later in the day. Interestingly urgency was a major risk factor for MACE.

Authors’ Feedback: We decided to focus on patients’ characteristics and clinical features to make our results more significant to clinical practice. Unfortunately, we didn’t have access to reasons for delays and operating issues. Additionally, we agree that this should be discussed as a limitation. We added the following in our manuscript text.

Discussion (page 14, line 303): “Not only patients’ characteristics and clinical aspects are responsible for time distribution; operating issues and surgeries delay could affect the schedule of the surgeries, which were not assessed in our study.”

8. Reviewer: The pattern of referral to this tertiary hospital was mentioned - as a possible system factor. Regional referral patterns for vascular surgery are well recognised, which may be related to availability of vascular surgery specialists and appropriate support services. If not in the data collected can the authors please comment on the factors in their hospital/region that influence what they have reported which could lead to outcome improvement.

Authors’ Feedback: We thank the reviewer for the comment. Our Institution is a tertiary University Hospital that deals with more complex surgical interventions like arterial vascular surgeries. In the Brazilian public health system, more complex interventions like those described in this study can occur only in specialized Hospitals. 100% of the patients in the present study are included in the Brazilian public health system.

9. Reviewer: Discussion page 12 line 271 it says “Additionally, our population were exclusive of patients undergoing arterial vascular surgery, which are patients at greater cardiac risk undergoing mostly high-risk operations. -could the authors please explain ? Do they mean Additionally, our population included only patients... Or Additionally, our population were exclusively patients undergoing…

Authors’ Feedback: We have corrected the sentence, so it is better represented.

Discussion (page 13, line 284): “We did not evaluate the incidence of MACE at long term follow-up, thus we could have missed some events. Also, since we had a higher-risk population with patients undergoing vascular surgeries that need cardiac referral, the elevated all-cause mortality rates observed may have overshadow other cardiac events.”

10. Reviewer: The Graphical abstract fig 2 should be revisited as it misses the point of the study – This reviewer would suggest that the study is not primarily about “need for better postoperative outcomes control” (not sure what that means). The message is about operations done later in the day – How can we insure the high risk patients are better prepared and more often operated earlier in the day.

Authors’ Feedback: We agree with the comment made by the reviewer. Therefore, we have updated our Fig 2 rephrasing “need for better postoperative outcomes control” to “Influence of surgery starting time on mortality and cardiac events”.

11. Reviewer: Similarly the Conclusion should be revisited – Vascular surgery patients operated later in the day are sicker, more urgent and have worse outcomes with double the mortality at 30m days and even out to one year. Steps are needed to improve outcomes which may include starting operations earlier.

Authors’ Feedback: We thank the reviewer for the comment but, in our opinion, the conclusion suits and responds our study objectives. The study design and the statistical analysis showed that vascular surgeries performed in the afternoon/night were independently associated with higher MACE and mortality in 30 days and one year. As suggested by the reviewer, we stated at the end of discussion section that prioritizing high-risk patients for operating during the morning may be a strategy for reducing cardiovascular complications.

---

## [Decision Letter · Decision Letter 1]

8 Jul 2022

PONE-D-21-21807R1Impact of the period of the day on all-cause mortality and major cardiovascular complications after arterial vascular surgeriesPLOS ONE

Dear Dr. Caramelli,

Thank you for submitting your revised manuscript to PLOS ONE. After careful consideration, we feel that it has merit but does not fully meet PLOS ONE’s publication criteria as it currently stands. Therefore, we invite you to submit a revised version of the manuscript that addresses the points raised during the review process.

Your manuscript has been reassessed by the two reviewers from the previous round, whose reports can be found below. As you will see from the comments, the reviewers acknowledge that the manuscript has improved, but there remain significant concerns (outlined by reviewer 2) that the manuscript does not clearly communicate the contribution your study makes to the body of academic knowledge. Please ensure you respond to each point carefully in your response to reviewers document, and modify your manuscript accordingly.

We look forward to receiving your revised manuscript.

Kind regards,

Joseph Donlan

Editorial Office

PLOS ONE

Reviewers' comments:

Reviewer's Responses to Questions

**Comments to the Author**

1. If the authors have adequately addressed your comments raised in a previous round of review and you feel that this manuscript is now acceptable for publication, you may indicate that here to bypass the “Comments to the Author” section, enter your conflict of interest statement in the “Confidential to Editor” section, and submit your "Accept" recommendation.

Reviewer #1: All comments have been addressed

Reviewer #2: (No Response)

2. Is the manuscript technically sound, and do the data support the conclusions?

Reviewer #1: Yes

Reviewer #2: Partly

3. Has the statistical analysis been performed appropriately and rigorously? 

Reviewer #1: Yes

Reviewer #2: Yes

4. Have the authors made all data underlying the findings in their manuscript fully available?

Reviewer #1: Yes

Reviewer #2: Yes

5. Is the manuscript presented in an intelligible fashion and written in standard English?

Reviewer #1: Yes

Reviewer #2: Yes

6. Review Comments to the Author

Reviewer #1: The remarks were taken into account appropriately. The few points that could not be corrected, due to the internal functioning of their hospital, were justified.

Reviewer #2: 1.Thank you and Appreciation to the authors for the attention given to the reviewers suggestions. The statistical tools used are well defended. This reviewer still has concern that the authors do not use this data to say something new or give strength to ways of dealing with the problem. The conclusion after all is not new - we all know about this phenomena.

The following comments are suggestions towards achieving this

2.The benefit of this paper to influence practice – while authors argue they wanted to have a clinical / clinician focus it

Assumes this will lead to change – but we have known for a long time that urgent procedures occur more often in the later part of the day with sicker people , reduced supporting services etc . Can the authors contribute more critically? If cardiology work up is so important – shouldn’t that be made more available – line 100 and 110. If this is true why not explore this in the data

3.Along the same lines in the method it says “This is part of a prospective cohort registry including consecutive patients undergoing non-cardiac arterial vascular surgery between 2012 and 2018 at Hospital das Clínicas, University of São Paulo Medical School, São Paulo, Brazil, for whom a preoperative cardiologic consultation was requested”. What does this mean – when you have included urgent patients who didn’t get this consultation. Your addition line 99-100 makes this confusing. Perhaps you could rewrite the description of your cohort

4 Is it not of interest to know who missed out - as requested in the previous peer review who were these and what the role of your preoperative assessment achieved for those who did get it ? Otherwise there is little relevance of mentioning preoperative consultation to the outcome of the study .

5.Reference to reoperation in previous review assumed the patients and the index procedure were only counted once but rather that reoperation has other possible implications – when reoperation is the index case, and that subsequent reoperation would be an important factor in outcomes at 30 days and one year.

6. Line 328 Not sure what the addition (redline version) means “We may have missed patients” ? see the confusion in comment 3 above. Does this mean all the patients (including the urgents) had vascular surgeon’s request made for cardiology consultation (and not all got it) and then there were a few for whom the vascular team didn’t get around to it and they are not in the study population ?

7. This comes back to what is the relevance of the consultation why is cardiac consultation important. It seems to this reviewer that the authors have alluded to this in the last paragraph but unfortunately have not used the data they have to explore this e.g there presumably are at least 1000 subjects who were referred for the cardiology consultation -

Another example you present data of differences in cardiac drugs - surely a benefit of seeing a cardiologist which indicates urgent cases seem not to be protected as well etc

Another example might be :Cardiac risk Index – speak of risk – can we presume this data is more likely to be documented by cardiology consultation ? This reviewer suspects this is far more important than fasting time !

8. Thankyou for the additional table This is quite revealing – amputations left to the end of the day , maybe for the juniors, often very sick, AAAE mainly during the morning – good idea , LLR being done late in the day . This deserves comment

7. PLOS authors have the option to publish the peer review history of their article (what does this mean?). If published, this will include your full peer review and any attached files.

Reviewer #1: No

Reviewer #2: No

---

## [Author Response · Author response to Decision Letter 1]

27 Jul 2022

Response to Reviewers

Impact of the period of the day on all-cause mortality and major cardiovascular complications after arterial vascular surgeries

Dear Editor and Reviewers,

The first round of Review was really enriching for us. We appreciate your consideration on reviewing our manuscript. We have carefully read and analyzed your comments and provided answers and changes when necessary.

For this second round of Review, please note that changes we made in the manuscript text are marked as the “red” color. Also, we are providing the files of both marked and clean “Impact of the period of the day on all-cause mortality and major cardiovascular complications after arterial vascular surgeries” reviewed versions attached.

1. If the authors have adequately addressed your comments raised in a previous round of review and you feel that this manuscript is now acceptable for publication, you may indicate that here to bypass the “Comments to the Author” section, enter your conflict of interest statement in the “Confidential to Editor” section, and submit your "Accept" recommendation.

Reviewer #1: All comments have been addressed

Reviewer #2: (No Response)

2. Is the manuscript technically sound, and do the data support the conclusions?

Reviewer #1: Yes

Reviewer #2: Partly

3. Has the statistical analysis been performed appropriately and rigorously?

Reviewer #1: Yes

Reviewer #2: Yes

4. Have the authors made all data underlying the findings in their manuscript fully available?

Reviewer #1: Yes

Reviewer #2: Yes

5. Is the manuscript presented in an intelligible fashion and written in standard English?

Reviewer #1: Yes

Reviewer #2: Yes

6. Review Comments to the Author

List of Changes – Reviewers’ Comments

Reviewer # 2

1. Reviewer: Thank you and Appreciation to the authors for the attention given to the reviewers’ suggestions. The statistical tools used are well defended. This reviewer still has concern that the authors do not use this data to say something new or give strength to ways of dealing with the problem. The conclusion after all is not new - we all know about this phenomenon. The following comments are suggestions towards achieving this.

Authors’ Feedback: We thank the reviewer for the comments. In the following paragraphs, we will address all issues indicated by the Reviewer #2 and make changes to the manuscript accordingly.

2. Reviewer: The benefit of this paper to influence practice – while authors argue they wanted to have a clinical / clinician focus it. Assumes this will lead to change – but we have known for a long time that urgent procedures occur more often in the later part of the day with sicker people, reduced supporting services etc. Can the authors contribute more critically? If cardiology work up is so important – shouldn’t that be made more available – line 100 and 110. If this is true, why not explore this in the data.

Authors’ Feedback: Thanks for the suggestions. Addressing issues raised by the Reviewer in comments 2 and 3, we have rewritten the “Study Overview and Population” paragraph so its clearer and better understandable. Changes were provided in the final version of the manuscript.

Materials and Methods (page 4, line 97): “Cardiology consultation and perioperative evaluation is provided in less than 24 hours in our hospital. For this study, we excluded patients from whom it was not possible to perform perioperative cardiovascular evaluation or to obtain time of the surgery with the available documentation. We included patients undergoing all arterial vascular surgeries: open or endovascular (for aorta, peripheral artery, visceral artery, and carotid artery diseases and amputations due to limb ischemia), emergent, urgent, or elective. Of note, in the present study, urgent/emergency procedures were considered interventions for acute onset pathologies or clinical deterioration without previously schedule or outpatient evaluation that were able to receive perioperative cardiac evaluation.”

3. Reviewer: Along the same lines in the method it says “This is part of a prospective cohort registry including consecutive patients undergoing non-cardiac arterial vascular surgery between 2012 and 2018 at Hospital das Clínicas, University of São Paulo Medical School, São Paulo, Brazil, for whom a preoperative cardiologic consultation was requested”. What does this mean – when you have included urgent patients who didn’t get this consultation. Your addition line 99-100 makes this confusing. Perhaps you could rewrite the description of your cohort.

Authors’ Feedback: As stated in Item 2, changes were provided in the final version of the manuscript.

4. Reviewer: Is it not of interest to know who missed out - as requested in the previous peer review who were these and what the role of your preoperative assessment achieved for those who did get it? Otherwise there is little relevance of mentioning preoperative consultation to the outcome of the study.

Authors’ Feedback: We appreciate the commentary. We made changes that can be seen in final version of the manuscript.

Discussion (page 14, line 330): “In clinical practice, low risk patients according to anesthesiology evaluation are often directly sent to surgical theater without further consultation. Given the nature of the present study (retrospective analysis of prospective collected data) we don’t have the number of patients operated without cardiac consultation, but we estimate a small number of cases that could not influence the final results of the present study.”

5. Reviewer: Reference to reoperation in previous review assumed the patients and the index procedure were only counted once but rather that reoperation has other possible implications – when reoperation is the index case, and that subsequent reoperation would be an important factor in outcomes at 30 days and one year.

Authors’ Feedback: We agree with reviewer that reoperation could be, per se, a risk factor for complications. However, this is not the purpose of the present study, that addresses the period of the day as a potential marker of complications. Kindly answering to Reviewer #2, we looked for reoperations in our database and found only 57 patients (4.4%) that were reoperated in one year after the first surgery.

6. Reviewer: Line 328 Not sure what the addition (redline version) means “We may have missed patients”? see the confusion in comment 3 above. Does this mean all the patients (including the urgents) had vascular surgeon’s request made for cardiology consultation (and not all got it) and then there were a few for whom the vascular team didn’t get around to it and they are not in the study population?

Authors’ Feedback: We have addressed to this issue in Itens 2 and 3. All patients included in our study had cardiology consultation and preoperative risk evaluated, including urgent/emergency surgeries. We did not include in our study patients in low risk strata, e.g., those referred to anesthesiology evaluation and directly sent to surgical theater without further consultation. We also addressed to this issue in Item 4.

7. Reviewer: This comes back to what is the relevance of the consultation why is cardiac consultation important. It seems to this reviewer that the authors have alluded to this in the last paragraph but unfortunately have not used the data they have to explore this e.g there presumably are at least 1000 subjects who were referred for the cardiology consultation – Another example you present data of differences in cardiac drugs - surely a benefit of seeing a cardiologist which indicates urgent cases seem not to be protected as well etc. Another example might be: Cardiac risk Index – speak of risk – can we presume this data is more likely to be documented by cardiology consultation? This reviewer suspects this is far more important than fasting time!

Authors’ Feedback: We thank the reviewer for the comment. The main outcome of the present study is the influence of the period of the day in the occurrence of complications and not the effect of cardiac consultation. As matter of fact, having cardiac consultation was an inclusion criteria to the study The role of cardiac consultation in this setting is also an important question that can be addressed in future studies.

8. Reviewer: Thank you for the additional table This is quite revealing – amputations left to the end of the day , maybe for the juniors, often very sick, AAAE mainly during the morning – good idea , LLR being done late in the day . This deserves comment.

Authors’ Feedback: We agree that showing this data is important for readers, but there is no power in our study to get conclusions or get clinical recommendations for each type of interventions separately, once we have not previously designed our study for those outcomes.

---

## [Decision Letter · Decision Letter 2]

20 Sep 2022

PONE-D-21-21807R2Impact of the period of the day on all-cause mortality and major cardiovascular complications after arterial vascular surgeriesPLOS ONE

Dear Dr. Caramelli,

Thank you for submitting your manuscript to PLOS ONE. After careful consideration, we feel that it has merit but does not fully meet PLOS ONE’s publication criteria as it currently stands. Therefore, we invite you to submit a revised version of the manuscript that addresses the points raised during the review process.

Please revise.

We look forward to receiving your revised manuscript.

Kind regards,

Academic Editor

PLOS ONE

Reviewers' comments:

Reviewer's Responses to Questions

**Comments to the Author**

1. If the authors have adequately addressed your comments raised in a previous round of review and you feel that this manuscript is now acceptable for publication, you may indicate that here to bypass the “Comments to the Author” section, enter your conflict of interest statement in the “Confidential to Editor” section, and submit your "Accept" recommendation.

Reviewer #1: All comments have been addressed

Reviewer #2: (No Response)

2. Is the manuscript technically sound, and do the data support the conclusions?

Reviewer #1: Yes

Reviewer #2: No

3. Has the statistical analysis been performed appropriately and rigorously? 

Reviewer #1: Yes

Reviewer #2: Yes

4. Have the authors made all data underlying the findings in their manuscript fully available?

Reviewer #1: Yes

Reviewer #2: Yes

5. Is the manuscript presented in an intelligible fashion and written in standard English?

Reviewer #1: Yes

Reviewer #2: Yes

6. Review Comments to the Author

Reviewer #1: (No Response)

Reviewer #2: The account of who did and who didnt get cardiology consultation is at best confusing. If a patient with a rupture AAA appeared at 11 pm was cardiology consultation available or not prior to surgery ? What happened to all the patients for whom the anaesthetist made decisions. Why could you not previously account for who did and who did not have consultation when you now say only those with consultation were included in the study ?

Please describe your consultation service that sees all patients within 24 hours - what hours of the day is the service not available . Why the insistence to describe this service and use it as your selection criteria and not want to explore its relevance?

7. PLOS authors have the option to publish the peer review history of their article (what does this mean?). If published, this will include your full peer review and any attached files.

Reviewer #1: No

Reviewer #2: No

---

## [Author Response · Author response to Decision Letter 2]

24 Sep 2022

Dear Editor and Reviewers,

The first round of Review was really enriching for us. We appreciate your consideration on reviewing our manuscript. We have carefully read and analyzed your comments and provided answers and changes when necessary.

For this second round of Review, please note that changes we made in the manuscript text are marked as the “red” color. Also, we are providing the files of both marked and clean “Impact of the period of the day on all-cause mortality and major cardiovascular complications after arterial vascular surgeries” reviewed versions attached.

1. If the authors have adequately addressed your comments raised in a previous round of review and you feel that this manuscript is now acceptable for publication, you may indicate that here to bypass the “Comments to the Author” section, enter your conflict of interest statement in the “Confidential to Editor” section, and submit your "Accept" recommendation.

Reviewer #1: All comments have been addressed

Reviewer #2: (No Response)

2. Is the manuscript technically sound, and do the data support the conclusions?

Reviewer #1: Yes

Reviewer #2: Partly

3. Has the statistical analysis been performed appropriately and rigorously?

Reviewer #1: Yes

Reviewer #2: Yes

4. Have the authors made all data underlying the findings in their manuscript fully available?

Reviewer #1: Yes

Reviewer #2: Yes

5. Is the manuscript presented in an intelligible fashion and written in standard English?

Reviewer #1: Yes

Reviewer #2: Yes

---

## [Decision Letter · Decision Letter 3]

17 Oct 2022

PONE-D-21-21807R3Impact of the period of the day on all-cause mortality and major cardiovascular complications after arterial vascular surgeriesPLOS ONE

Dear Dr. Caramelli,

Thank you for submitting your manuscript to PLOS ONE. After careful consideration, we feel that it has merit but does not fully meet PLOS ONE’s publication criteria as it currently stands. Therefore, we invite you to submit a revised version of the manuscript that addresses the points raised during the review process.

Please revise. 

We look forward to receiving your revised manuscript.

Kind regards,

Academic Editor

PLOS ONE

Reviewers' comments:

Reviewer's Responses to Questions

**Comments to the Author**

1. If the authors have adequately addressed your comments raised in a previous round of review and you feel that this manuscript is now acceptable for publication, you may indicate that here to bypass the “Comments to the Author” section, enter your conflict of interest statement in the “Confidential to Editor” section, and submit your "Accept" recommendation.

Reviewer #3: (No Response)

Reviewer #4: All comments have been addressed

2. Is the manuscript technically sound, and do the data support the conclusions?

Reviewer #3: Partly

Reviewer #4: Yes

3. Has the statistical analysis been performed appropriately and rigorously? 

Reviewer #3: Yes

Reviewer #4: Yes

4. Have the authors made all data underlying the findings in their manuscript fully available?

Reviewer #3: Yes

Reviewer #4: Yes

5. Is the manuscript presented in an intelligible fashion and written in standard English?

Reviewer #3: Yes

Reviewer #4: Yes

6. Review Comments to the Author

Reviewer #3: A very major flaw with this study is that there is no analysis made of “afternoon” surgery (ie normal working hours) and out of hours (emergent) surgery. Out of hours surgery will always have poorer outcomes given the emergent nature of the work.

Reviewer #4: (No Response)

7. PLOS authors have the option to publish the peer review history of their article (what does this mean?). If published, this will include your full peer review and any attached files.

Reviewer #3: **Yes: **Sandeep Bahia

Reviewer #4: No

---

## [Author Response · Author response to Decision Letter 3]

30 Nov 2022

Dear Editor and Reviewers,

We appreciate your consideration on reviewing our manuscript. We have carefully read and analyzed your comments and provided answers and changes when necessary.

For this third round of Review, please note that changes we made in the manuscript text are marked as the “red” color. Also, we are providing the files of both marked and clean “Impact of the period of the day on all-cause mortality and major cardiovascular complications after arterial vascular surgeries” reviewed versions attached.

1. If the authors have adequately addressed your comments raised in a previous round of review and you feel that this manuscript is now acceptable for publication, you may indicate that here to bypass the “Comments to the Author” section, enter your conflict of interest statement in the “Confidential to Editor” section, and submit your "Accept" recommendation.

Reviewer #3: (No Response)

Reviewer #4: All comments have been addressed

2. Is the manuscript technically sound, and do the data support the conclusions?

Reviewer #3: Partly

Reviewer #4: Yes

3. Has the statistical analysis been performed appropriately and rigorously? 

Reviewer #3: Yes

Reviewer #4: Yes

4. Have the authors made all data underlying the findings in their manuscript fully available?

Reviewer #3: Yes

Reviewer #4: Yes

5. Is the manuscript presented in an intelligible fashion and written in standard English?

Reviewer #3: Yes

Reviewer #4: Yes

6. Review Comments to the Author. Please use the space provided to explain your answers to the questions above. You may also include additional comments for the author, including concerns about dual publication, research ethics, or publication ethics. (Please upload your review as an attachment if it exceeds 20,000 characters)

Reviewer #3: A very major flaw with this study is that there is no analysis made of “afternoon” surgery (ie normal working hours) and out of hours (emergent) surgery. Out of hours surgery will always have poorer outcomes given the emergent nature of the work.

Reviewer #4: (No Response)

List of Changes – Reviewers’ Comments

Reviewer # 3

1. Reviewer: A very major flaw with this study is that there is no analysis made of “afternoon” surgery (ie normal working hours) and out of hours (emergent) surgery. Out of hours surgery will always have poorer outcomes given the emergent nature of the work.

Authors’ Feedback: We thank the reviewer for the comment. In the following paragraphs, we will address the collinearity test we made analyzing the endpoints in the urgent/emergency surgeries group. We made changes to the manuscript accordingly.

Materials and Methods (page 7, line 188): “Sensitivity Analysis - Since urgent/emergency surgeries are more frequent in the night period they could be correlated variables. To avoid confounding results a sensitivity analysis was made considering only the population undergoing urgent/emergency surgery, submitting them to the same statistical analysis performed in the main cohort. ”

Results (page 12, line 264): “Of the total, 478 patients underwent urgent/emergency surgery: 181 (37.9%) during the morning and 297 (62.1%) during the afternoon/night period. Considering only urgent/emergency surgery population, significant increase in all-cause mortality in the afternoon/evening surgery group was observed after 30 days (HR 2.57 [95% CI 1.69 - 3.93], P value < 0.001) and one year (HR 2 .43 [95% CI 1.75 - 3.37], P value < 0.001). After multivariate analysis, 30-day all-cause mortality was higher in the afternoon/night period (aHR 2.43 [95%CI 1.54 - 3.83], P = < 0.001), as well as one year mortality (aHR 2.35 [95%CI 1.65 – 3.33], P =< 0.001 – S3 Fig. Mortality in Urgent/Emergency Surgeries).

The incidence of PMI in the urgent/emergency surgery group was also higher in patients operated in the afternoon/evening period: 44.2% versus 33.0%, respectively (OR 1.82 [95%CI 1.22 - 2.71], P- value = 0.003). In the multivariate analysis, urgent/emergency surgeries performed in the afternoon/evening presented independent association with the incidence of PMI (aHR 1.4 [95%CI 1.03 – 1.91], P =0.03). After 30 days, the incidence of MACE in urgent/emergency surgery group was higher in the afternoon/evening surgery population. Surgeries performed in the afternoon/evening period were independently associated with increased MACE after correction for confounding factors in the multivariable analysis (aHR 1.51 [95%CI 1.10 – 2.06], P =0.010).”

Supporting Material (page 21, line 526): S3 Fig. Mortality in Urgent/Emergency Surgeries

---

## [Decision Letter · Decision Letter 4]

19 Dec 2022

Impact of the period of the day on all-cause mortality and major cardiovascular complications after arterial vascular surgeries

PONE-D-21-21807R4

Dear Dr. Caramelli,

We’re pleased to inform you that your manuscript has been judged scientifically suitable for publication and will be formally accepted for publication once it meets all outstanding technical requirements.

Kind regards,

Academic Editor

PLOS ONE

Additional Editor Comments (optional):

Reviewers' comments:

Reviewer's Responses to Questions

**Comments to the Author**

1. If the authors have adequately addressed your comments raised in a previous round of review and you feel that this manuscript is now acceptable for publication, you may indicate that here to bypass the “Comments to the Author” section, enter your conflict of interest statement in the “Confidential to Editor” section, and submit your "Accept" recommendation.

Reviewer #4: All comments have been addressed

Reviewer #5: All comments have been addressed

2. Is the manuscript technically sound, and do the data support the conclusions?

Reviewer #4: Yes

Reviewer #5: Yes

3. Has the statistical analysis been performed appropriately and rigorously? 

Reviewer #4: Yes

Reviewer #5: Yes

4. Have the authors made all data underlying the findings in their manuscript fully available?

Reviewer #4: Yes

Reviewer #5: Yes

5. Is the manuscript presented in an intelligible fashion and written in standard English?

Reviewer #4: Yes

Reviewer #5: No

6. Review Comments to the Author

Reviewer #4: (No Response)

Reviewer #5: Logic outcomes and useful for the general vascular surgeons. It describes a very important problem that faces us.

7. PLOS authors have the option to publish the peer review history of their article (what does this mean?). If published, this will include your full peer review and any attached files.

Reviewer #4: No

Reviewer #5: **Yes: **Aram Baram

---

## [Editor Report · Acceptance letter]

22 Dec 2022

PONE-D-21-21807R4 

Impact of the period of the day on all-cause mortality and major cardiovascular complications after arterial vascular surgeries 

Dear Dr. Caramelli:

I'm pleased to inform you that your manuscript has been deemed suitable for publication in PLOS ONE. Congratulations! Your manuscript is now with our production department. 

Kind regards, 

on behalf of

Dr. Robert Jeenchen Chen 

Academic Editor

PLOS ONE